# Consistent Characterization of Color Degradation Due to Artificial Aging Procedures at Popular Pigments of Byzantine Iconography

**Stamatios Amanatiadis** [1,*] **, Georgios Apostolidis** [1,2] **and Georgios Karagiannis** [1,2]

1    "Ormylia" Foundation, Art Diagnosis Center, GR-63071 Ormylia, Greece; ormylia@artdiagnosis.gr (G.A.); info@diagnosismultisystems.eu (G.K.)
2    Diagnosis Multisystems IKE, GR-63200 Nea Moudania, Greece
*    Correspondence: amanatiadis@artdiagnosis.gr

**Abstract:** The degradation effects of artificial aging on the "true" pigment color of Byzantine iconography are thoroughly investigated in this work. For this purpose, a multi-material palette is fabricated, consisting of various popular egg-tempera pigments, while the original recipes from the literature are utilized in order to mimic the genuine art of Byzantine painters. Then, artificial aging procedures are appropriately employed to simulate environmental fluctuations in historical buildings, such as churches. A total of four time steps are investigated, including the initial condition, and pigments' spectra in the ultraviolet/visible (UV/Vis) area are acquired in the diffuse reflectance mode at each individual step. Moreover, a color characterization procedure is realized via the quantification of lightness and saturation by means of the measured UV/Vis spectrum. The main objectives of this work are to determine the color stability, the type of color degradation, and generally the color response through time of the studied pigments. The extracted results indicate that a couple of pigments suffer severe color degradation while the majority present moderate darkening or discoloration.

**Keywords:** aging; Byzantine art; cultural heritage; egg-tempera; lightness; saturation; UV/Vis

## 1. Introduction

During the recent history of cultural heritage science, various techniques from physics, chemistry, and engineering have been utilized for the identification of the material and chemical composition of artworks [1–8]. One of the most significant class of non-destructive modalities is spectroscopy that evaluates the material's spectrum due to electromagnetic interaction utilizing various modes, namely reflection, emission, or absorption [9]. Several subcategories are identified with respect to the frequency region of interest, such as the popular infrared spectroscopy that provides chemical structure information via the resonant frequencies of the molecules. An additional measurement technique is ultraviolet/visible (UV/Vis) spectroscopy, which focuses on the corresponding frequency regime. The importance of the latter is the fact that the visible range is the main perception of electromagnetic radiation for a human through the eyes. Consequently, the post-processing of the acquired UV/Vis spectrum leads in the perceived color that constitutes the initial interaction of a human with artworks [10–13]. The majority of the cultural heritage assets are dated centuries ago, and the physiochemical changes of the artwork's pigments due to aging tend to considerably alter the color perception compared to the initial stage. In particular, Byzantine icons suffer severe degradation due to environmental fluctuations in not heated/conditioned historical buildings, such as churches. As a consequence, the characterization of color degradation due to aging procedures is a critical objective that facilitates the appropriate conservation task of artworks. Moreover, simulation operations can be performed to predict the initial coloring of iconic artworks via degradation inversion towards the past.

In many cases, the durability of materials and their behavior are tested under natural aging conditions by exposing them outdoors or indoors in the proper environment where they are supposed to be in their lifetime. However, it is often necessary to more rapidly determine the effects of light, heat, and moisture on the physical, chemical, and optical properties of materials with artificially accelerated weathering or artificial accelerated irradiation exposures that use specific laboratory light sources. The exposure of materials in laboratory aging devices is conducted in more controlled conditions than those found in natural conditions and are aimed at both studying the effect of single weathering factors and accelerating material degradation. Although no single laboratory exposure test can be considered a total simulation of natural exposures [14–16], the main factors that cause weather degradation can be properly adjusted to conduct an approximation in common sense. The aforementioned main factors are light, heat, and moisture, and they can be used to describe the exposure conditions in a broad range [17]. The main guidelines for the application of artificial aging for painting materials are found in two basic ISO standards, i.e., ISO16474-1:2013 [18] and ISO16474-2:2013 [19].

Aging encompasses a set of physiochemical processes that paintings undergo within months to years of their realization. Initially, the paint, which is a mixture of pigments in a binder or a mixture of binders, undergoes drying, which leads to the formation of a polymeric matrix. Longer-term aging leads to various different phenomena, and these have been studied extensively on easel and panel paintings [20–24]. In the museum field, photochemical damage is often assessed by measuring the surface color change resulting from light exposure. The data on the rate of fading caused by a particular light source over a relatively long time period were extrapolated to suggest a rough order-of-magnitude extent of change over the shorter time of exposure to photoflash. This approach builds on the assumption that the reciprocity principle is applicable. Most inorganic pigments are acknowledged as being very light stable, with a few well-known exceptions that include vermilion, lead (II) oxide, lead chromate, Prussian blues, and some copper greens [25]. Moreover, there are several studies that focus on the color characterization due to artificial aging degradation for a specific group of pigments witnessing that the various physiochemical changes are able to significantly alter the overall perception of a color [26–28]. Concerning Byzantine iconography, there is a couple studies that evaluate the physiochemical degradation type using a multimodal approach [29,30].

The purpose of the current work is to examine and characterize the color degradation during artificial aging of egg-tempera pure pigments used in Byzantine art. In particular, the objectives of this paper are not limited in the study of the color stability and the type of degradation but are extended towards the determination of color response through time, namely the examination of whether a degradation is observed at early or later exposure stages. For this reason, a multi-material palette is fabricated that includes various popular pigments of this specific era. The analysis is focused on the pure pigments, namely without the presence of protective varnishes, lightings, or writings to avoid complex discoloration parameters that can lead in arbitrary conclusions. Moreover, a realistic artificial aging procedure of four steps is conducted through the simulation of temperature, humidity, and light of historical buildings, in which Byzantine artworks are commonly conserved, namely churches. The acquisition of UV/Vis spectrum measurements is realized at every individual time step and a quantification procedure is performed, regarding pigment color, via a proposed scheme for the determination of lightness and saturation through the rich in information spectra. Moreover, a CIELAB characterization is realized as a complementary procedure to evaluate the color difference of pigments. The extracted results indicate that the majority of the pigments are moderately influenced by aging, mainly darkened or discolored, while a severe degradation is observed to a couple of them.

### 1.1. Examined Historical Pigments

The color of a pigment is determined, mainly via the refractive index along an absorption region. A pigment must have a large refractive index to be effective as an attributed

color. Typically, the refractive index is bigger in the region of large wavelengths and lower in the region of short wavelengths. For this reason, the red, orange, and yellow pigments commonly have higher refractive index than the green, blue, and violet ones. Additionally, the size and the shape of the grains of the pigment also play important roles in the performance of the painting. Generally, the granules must be sufficiently small and uniform to allow their uniform dispersion within the binder and to facilitate the spread of paste by brush. Certain colors, such as azurite and malachite, are the exception since a richer color is achieved when the size of the grains is bigger.

Generally, the grains of the mineral pigments are broken with irregular and angular shapes and their size varies depending on the toughness of the mineral and the method of pulverization. They are usually moderate to very coarse as in the case of azurite and cinnabar. The earth pigments derived from natural sediments consist of small granules of various sizes and are usually processed with the pulverization process to be separated. They have a heterogeneous composition and most comprehensive form as the green earth and sienna. Next, a more comprehensive analysis is presented concerning the utilized pigments [31–33].

1. Massicot, a type of yellow lead monoxide (PbO) that, according to ancient sources, such as Pliny, is known since antiquity. It is produced by slight firing lead white, which eliminates $CO_2$ and water, leaving behind a soft yellow powder with a sulfur color. Generally, it is not affected by light but, with prolonged exposure to moisture, can be reversed in white lead.
2. The warm yellow of Naples is the oldest artificially produced pigment, used from ancient times. Today, yellow of Naples is known as lead antimoniate, and it is produced artificially by the firing of lead oxide with antimony oxide or the salts of the two metals. The color ranges from light yellow to orange-yellow, depending on the ratio of the two components and the temperature of the preparation.

Generally, ochres are among the oldest pigments in the world, and their traces have been found in all historical periods. They were considered important colors in the Middle Ages and were in the palette with hues from light yellow to dark brown. The ochre is a natural earth color consisting of aluminosilicate materials and its color is due to iron oxide, either in a hydrated or anhydrous form.

3. The yellow ochre color owes its color to the presence of hydrated iron oxide, mainly in the goethite mineral. Apart from iron oxides, it may also contain impurities of plaster and magnesium carbonate.
4. On the other hand, red ochre consists of iron oxide in a substantially anhydrous form with aluminosilicate contaminants. The complexion varies according to the degree of hydration and the origin of minerals. Similar to every iron oxide, the red pigment is very stable and insensitive to light. Generally, the pigment tends to dry quickly.
5. Finally, warm ochre is a variant of yellow ochre with a warmer hue.
6. The mineral hematite is quite hard and compact and is almost composed of pure anhydrous iron oxide $Fe_2O_3$. The pigment is characterized by a rich dark color with purple-red hue. Microscopically, the granules differ from other ochres as they present a branched form of bright elongated alder.
7. The raw sienna is a special form of yellow ochre that got its name from the Tuscan region where the particularly beautiful shade of its color comes from. The rich warm maroon color of burnt Sienna is a result of heating under oxidizing conditions (calcining) the raw Sienna. By calcination, the hydrated iron oxide is converted to anhydrous iron oxide.
8. Although the mineral minium occurs quite frequently in nature, its use as a pigment in antiquity is questioned. It was often found as an illumination on cinnabar or as a thin decoration to create a glowing grid that resembles lace. Chemically, minium is a lead tetroxide ($Pb_3O_4$). According to some ancient recipes, the pigment is produced when metallic lead or its minerals are heated. Pliny calls it "the color of the flame"

and, indeed, the bright intense red color of minium is mostly characterized as orange rather than red.

9. The cinnabar is a compact heavy red mineral (HgS), which is the main mineral of mercury. The pigment is characterized by its rich cherry color, often located in the folds of clothing and dress of the Virgin as well as at the pink color of flesh. The cinnabar is considered sufficiently stable to light but sometimes tends to blacken, especially in egg-tempera.

10. The term green earth is applied to the earthy grey-green aluminosilicate minerals found in abundance all over the planet, and it is mainly composed of two related minerals: glauconite and celadonite. The color of the green earth minerals varies from grey-green to oily brown. Green earths are often described as the most permanent of all pigments, as they are not affected by atmospheric conditions or sunlight and as they do not react with solvent or other pigments.

11. The malachite mineral results from weathering copper ores, and it is found on the upper levels of copper ore deposits. In nature, malachite is associated with the rarer azurite, containing less chemically bound water. A particular feature of the malachite is that the performance of the color depends on the size of the particles. The richest intense green color is obtained from the coarse grains, while the fine species produces a pale green color.

12. The blue basic copper carbonate, azurite, is a natural pigment much more popular than the green counterpart, malachite. It is considered the most important blue pigment in European painting throughout the Middle Ages. Similar to malachite, its color depends on the grain size and it is, generally, unaffected by light. However, it can be darkened when heated or degrade to a green tint, while the pigment gradually losses its color over time, according to the literature.

13. No other dye was held in the highest regard in the Middle Ages other than the beautiful blue ultramarine. This rare and precious dye came from the natural mineral lapis lazuli, which was mainly, if not exclusively, found in the quarries of Badakhshan Province in Afghanistan. Ultramarine composition, essentially, includes Sodium (Na), aluminium (Al), sulfur (S), and silicon dioxide ($SiO_2$). In general, the strong and bright blue color is preserved when ultramarine is used with egg-tempera as a binder.

14. The cobalt blue in its pure form is a modern synthetic pigment discovered in 1802. It stands out in the blue color palette due to its strong cyan complexion and brightness compared to other blue pigments that tend to be darker and approach black when used in heavy layers, while it is stable in light.

15. Prussian blue is the first modern pigment that was introduced early in the 18th century. It is characterized by its strong coloring power using small concentrations of the pigment with any kind of binder. Characterized by the dark blue color with a slightly greenish glint, it tends to resemble black. The pigment has been described as both stable and unstable due to the variety of additives, methods of preparation, and choice of binder.

16. Another very important blue pigment used in the Middle Ages is the natural organic pigment indigo prepared from herbs of the Papilionaceae (*Indigofera tinctoria* L.) family, native to India. In Byzantine art, a special combination included an underpainting of indigo with thin coats of azurite as a lazure. As a pigment, it is sufficiently stable in light and air, and insoluble in water, alcohol, and ether but tends to discolor when exposed to ozone and atmospheric nitrogen dioxide.

## 2. Materials and Methods

### 2.1. Multi-Material Palette Details

A multi-material palette, depicted in Figure 1 has been designed and fabricated for the purpose of the pigment aging characterization in this work. According to the Byzantine iconography, successive layers of lightings and writings are applied by adding white lead or carbon black/red ochre, respectively, in the same mixture of initial color [34,35].

Although this procedure is adopted in various samples of our panel this paper focuses only on pure colors, namely without any lighting or writing. Moreover, a protective varnish is applied to the upper region of each sample, but the measurements are conducted only at unvarnished pigments. The main reason is that pigments with lighting, writing, or varnish are avoided originates from the complexity of the aging procedure. In particular, color degradation is investigated on pigments at their almost purest form, namely including only a binder, to minimize the effects of additional factors; thus, safer conclusions can be extracted.

Initially, a conventional plank plywood is selected since its role in color perception is not crucial due to the white preparation layer that covers it. Moreover, this specific wood is durable, avoiding severe deformations during the aging steps that can affect the color surface by creating cracks or even loss of material. The dimensions of the plank plywood are $45 \times 30.5\,\text{cm}^2$, comfortably fitting 60 samples of $4 \times 4\,\text{cm}^2$, while its thickness is 1.5 cm. Then, the wood is treated with a rabbit-skin glue layer before applying the preparation. The glue–water ratio is 1:12, i.e., 1 g glue powder in 12 mL of water. Finally, the preparation layer is composed of a more diluted glue mixture, 1:15, and hydrated plaster $CaSO_4 \cdot 2H_2O$ (Bologna gypsum) covered by four equally distributed layers of $CaCO_3$ chalk.

In the egg-tempera painting tradition, the binder is prepared with egg yolk and water. Specifically, the egg yolk is separated by the egg white and it is shuffled with the desired amount of water to create a suitable fluid emulsion, which is left for at least an hour to settle. There are no specific requirements regarding the ratio of color to binder. By nature, each pigment requires a certain percentage carrier in order to achieve a homogeneously thick paste having the respective absorptive capacity. Consequently, the colors are created with the appropriate concentration of each pigment in order to exploit its reflective ability. Moreover, it is intended to emulate the technique of Byzantine painters using original recipes from literature [34–39]. Note that Kremers Pigment Inc. provided the original studied pigments.

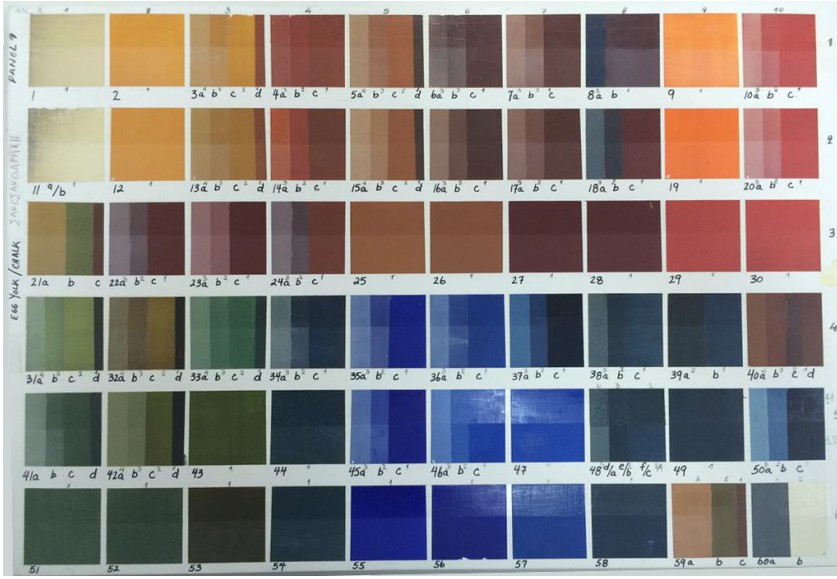

**Figure 1.** The multi-material palette of fundamental pigments of Byzantine iconography. A protective varnish is applied on the upper region of each pigment.

### 2.2. Artificial Aging Procedure

As aforementioned, there is a difficulty in the reproduction of the actual environmental conditions that affects material degradation in an aging chamber. As a consequence, it is not straightforward to set a good correlation between natural and artificial aging. However, in our work, an effort is conducted to simulate the environmental fluctuations in a not heated/conditioned historical building, such as a church, where Byzantine icons have been

and/or are stored for an extended time. In particular, the environmental conditions in the Mediterranean area throughout a year are emulated in terms of temperature, humidity, and light exposure considering that, in the summer, the temperature and sunshine are at the highest levels while humidity is moderate. The latter gets its highest values in autumn and spring, but the temperature and light exposure are decreased. Finally, in winter, the environmental parameters are at the lowest levels, except for humidity, which remains moderate.

In this work, an aging apparatus (Angelantoni ACS GTS60) equipped with a mercury vapor lamp HSW-400W E40 is utilized to account the chemical, visual, and mechanical deterioration induced by light, temperature, and humidity. Since the spectral emission of the mercury vapor lamp includes the lines listed in Table 1, a glass filter is used to cut shorter wavelengths (ordinary window glass passes about 90% of the light above 350 nm but blocks over 90% of the light below 300 nm).

**Table 1.** Emission lines of the vapor mercury lamp.

| Color | UVC | UVC | UVA | Violet | Blue | Green | Yellow |
|---|---|---|---|---|---|---|---|
| Wavelength (nm) | 184.45 | 253.7 | 365.4 | 404.7 | 435.8 | 546.1 | 578.2 |

The temperature and the humidity are decreased and increased periodically in order to simulate the seasons of the years within the time cycle of the 2 weeks. Temperature range between 48 °C and 60 °C and relative humidity between 40% and 60%, while the irradiance mean value is almost 800 W/m$^2$. As already noted, it is not accurate to quantify the artificial aging in terms of natural exposure, but it is intended to emulate approximately 50 years via each time cycle. Finally, a total amount of three aging cycles are performed, while a notation T1, T2, and T3 is used. The initial time-step, before aging cycles start, is notated as T0.

*2.3. Measurements at the Visible Spectrum and Quantification*

The evaluation of color changes in pigments due to artificial aging procedure requires the extraction of the "true" color. The latter is defined as the one that it is not influenced by illumination, viewing angle, and the observer's subjective perception. For this reason, UV/Vis spectroscopy is employed (a snapshot of the measurements is depicted in Figure 2), where UV and visible radiation only cause electronic excitation [40]. A monochromator (prism or grating) analyzes the white light in different monochromatic areas and selects the desired wavelength with high accuracy. In our case, a grating monochromator is utilized due to its advantageous linear performance at a wide UV/Vis spectral range. The detector detects the signal derived by the substance we want to determine, while the measurement or recording of the signal is achieved using a vulnerable photocell. The probe is assembled using an integration sphere of barium sulfate (BaSO$_4$) surface for in situ measurements of the diffuse reflectance radiation from the object. The integration sphere is the best approach as far as wavelength range and signal/noise ratio are concerned in order to provide the possibility for accurate measurements. The spot size of the probe is a circle with a diameter of approximately 2 mm.

The spectrometer used for the acquisition of the UV/Vis spectra is the Avantes AvaSpec of 200–1000 nm spectral region and 0.4–20 nm resolution, depending on the wavelength. The measurement procedure follows the standard ASTM E-903-96 [41]. Specifically, the objectivity of the color measurement is ensured via a calibration procedure of the maximum and minimum reflection values at the UV/Vis spectrum, utilizing white barium sulfate and carbon black, respectively. Finally, a quantification procedure is conducted in order to express numerically the color change. In this work, we attempted to extract standard quantification parameters, such as lightness and saturation, directly via the UV/Vis spectrum that includes the maximum information. Note that this is the reason that UV/Vis spectroscopy is employed rather than colorimetry, which is not able to

retrieve spectral details. Moreover, the conventional CIELAB color space is also utilized as a supporting post-processing procedure to the proposed, almost purely spectral, lightness, and saturation values.

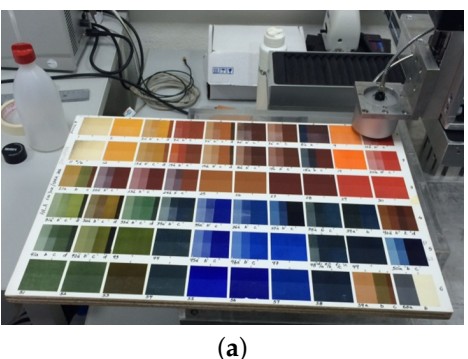

(**a**)

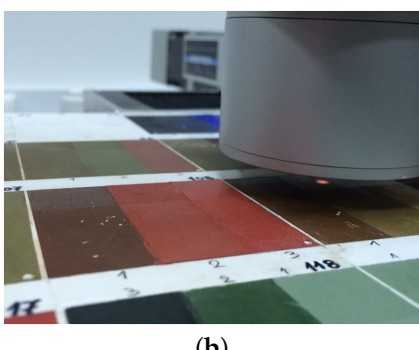

(**b**)

**Figure 2.** (**a**) Measurement of the egg tempera color panel and (**b**) a close-up of the measurement.

### 2.3.1. Lightness Evaluation from Visible Spectrum

The first important quantification parameter is the perceptual lightness of the pigments that is extracted directly through the UV/Vis measurement. Specifically, it is defined as the portion of the sample's spectrum intensity $s(\lambda)$ at the visible wavelength (400–700 nm) compared to the reference white $w(\lambda)$ and black $b(\lambda)$ ones, considered as the most luminous color via

$$L = \frac{\int_{400}^{700} [s(\lambda) - b(\lambda)]\mathrm{d}\lambda}{\int_{400}^{700} [w(\lambda) - b(\lambda)]\mathrm{d}\lambda}. \tag{1}$$

Consequently, the result of the lightness is a percentage, where the maximum (100%) and the minimum (0%) values correspond to the totally reflecting barium sulfate and totally absorbing carbon black, respectively. An example of the described procedure is briefly illustrated in Figure 3a for a cobalt blue sample, where the shaded part indicates the integration area.

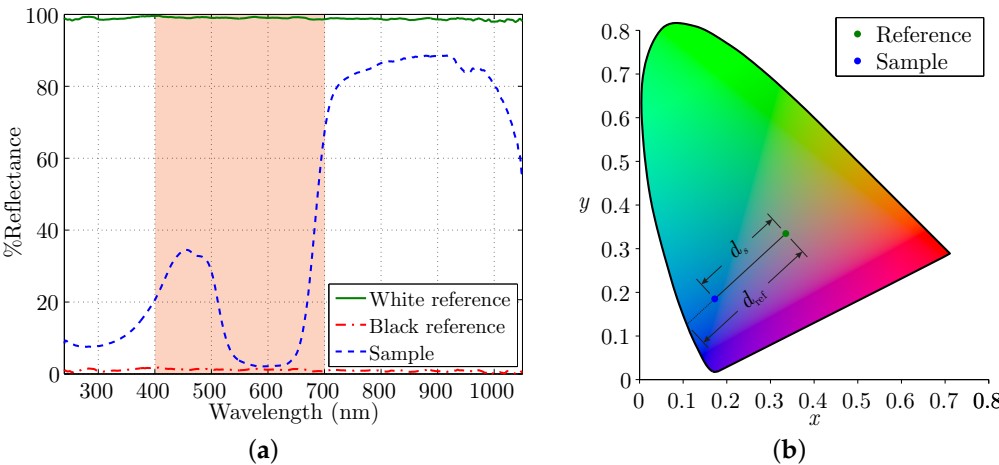

(**a**)　　　　　　　　　　　　　　　　　　(**b**)

**Figure 3.** Calculation procedure for (**a**) lightness and (**b**) saturation of a pigment's color.

### 2.3.2. Saturation Calculation via Tristimulus Values

Another significant parameter in color characterization is its saturation, which is determined by a combination of light intensity and how much it is distributed across the spectrum of different wavelengths. The purest (most saturated) color is achieved by using

just one wavelength at a high intensity, such as in laser light. Such colors are identified as spectral colors, and they are utilized for the saturation estimation of the pigments under investigation in this work.

Specifically, the colors that the human eye are able to perceive lie on the chromaticity diagram in Figure 3b. Note that this diagram is evaluated via the tristimulus values X, Y, and Z, an analogous system to the cone responses of the human eye. Moreover, the spectral colors, namely the most saturated ones, are at the diagram's border, while the absence of any color, namely the zero saturated gray, is approximately at the center {0.33, 0.33}. Finally, the saturation level of any other color is calculated via the portion of its distance $d_s$ to the equivalent spectral color one $d_{\mathrm{ref}}$ from gray.

### 2.3.3. Conventional CIELAB Color Space Representation

The measured spectra are also post-processed in order to evaluate the representation at the CIELAB color space that is considered perceptually uniform (a given numerical change corresponds to a similar perceived color change). It expresses color as three values: $L^*$ for perceptual lightness, and $a^*$ and $b^*$ for the four unique colors of human vision: red, green, blue, and yellow. These parameters are calculated via the aforementioned tristimulus values X, Y, and Z through the ISO/CIE 11664-4 standard [42]. It is worth mentioning that the CIELAB representation facilitates the color difference $\Delta E$ quantification between two colors with $L_1^*$, $a_1^*$, and $b_1^*$ and with $L_2^*$, $a_2^*$, and $b_2^*$ values via the formula

$$\Delta E = \sqrt{(L_2^* - L_1^*)^2 + (a_2^* - a_1^*)^2 + (b_2^* - b_1^*)^2}, \tag{2}$$

where a difference less than 2.3 is considered not noticeable.

### 3. Results and Discussion

The measurements are conducted for every pure pigment described in Section 1.1 for different aging time-step, including the initial condition. As already explained, the pigments with lightings, writings, or varnish are rejected to avoid the multi-parametric color degradation; therefore, more robust conclusions can be extracted. It is worth mentioning that an influence on the egg–yolk binder due to the aging procedure was expected. However, the chemical effect of the dicoloration is not in the scope of this work. In particular, the purpose of the paper is the characterization of egg-tempera pigment color change throughout equivalent time-cycles to indicate

- the overall color stability,
- whether the color degradation is observed at early exposure instances or later, and
- the type of color change in terms of lightness and saturation.

All of these objectives are related exclusively to color details, while the quantification is based on the rich in information UV/Vis spectrum rather than a colorimetric technique.

Some indicative UV/Vis spectra are depicted in Figure 4 for yellow (sample 2, Naples yellow), red (sample 9, cinnabar), green (sample 10, green earth), and blue (sample 14, cobalt blue). Initially, the spectrum of Naples yellow pigment (Figure 4a) is almost low at lower wavelengths, but it starts to rise after approximately 500 nm for any of the aging steps. However, there is a tendency towards overall lower values, especially at the last time-step T3. A similar behavior is observed for cinnabar (Figure 4b) since the spectrum is decreased for a progressing aging procedure, mainly after the second step T2. Moreover, the general increment is observed, now, at 600 nm, as expected due to the reddish hue.

Then, the green earth pigment (Figure 4c) is examined. In general, the differentiation due to the aging steps is not a significant one. This behavior is somewhat expected since it is considered the most permanent of colors. There is only a slight reduction in the overall intensity, while a maximum is detected at 550 nm, where the green hue is dominant. Finally, the cobalt blue pigment (Figure 4d) presents relatively different attributes compared to the previously discussed ones. In particular, the spectrum is now shifted towards increased

intensity values as the aging advances, primarily at lower wavelengths. Moreover, a local maximum is noted at the bluish hue of 350 nm, with a steep rise after 650 nm.

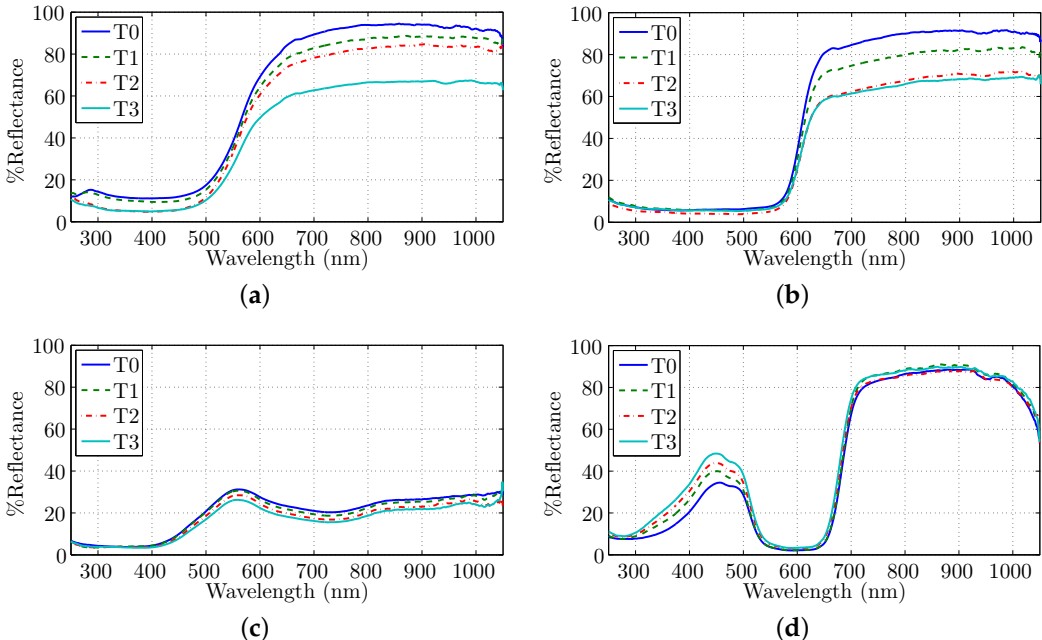

**Figure 4.** UV/Vis spectrum at the different aging steps for (**a**) yellow of Naples, (**b**) cinnabar, (**c**) green earth, and (**d**) cobalt blue pigments.

The spectral information provides some qualitative details concerning the aging influence on the color of the selected pigments. However, the visualization and quantification steps are required in order to comprehend the aging degradation. For this reason, the acquired spectra are processed to evaluate the colors at different time-steps as well as their lightness and saturation values via the methodologies described in Sections 2.3.1 and 2.3.2, respectively. Moreover, an auxiliary CIELAB characterization is performed for the determination of color difference, presented in Section 2.3.3.

Initially, the "true" color is extracted and illustrated in Figure 5. Here, there are four basic color categories regarding the alteration through time. The most interesting case includes massicot (sample 1) and minium (sample 8) since there is a severe degradation, especially after the second aging step. Additionally, the second category consists of Naples yellow (sample 2), yellow (sample 3) and warm (sample 5) ochres, burnt Sienna (sample 7), cinnabar (sample 9), and azurite (sample 12) since these pigments seem to darken as the aging progress. On the contrary, there is a class of pigments that tend to discolor, including ultramarine blue (sample 13), cobalt blue (sample 14), and indigo (sample 16). Finally, various pigments appear stable, regarding their color, particularly red ochre (sample 4), hematite (sample 6), green earth (sample 10), malachite (sample 11), and Prussian blue (sample 15).

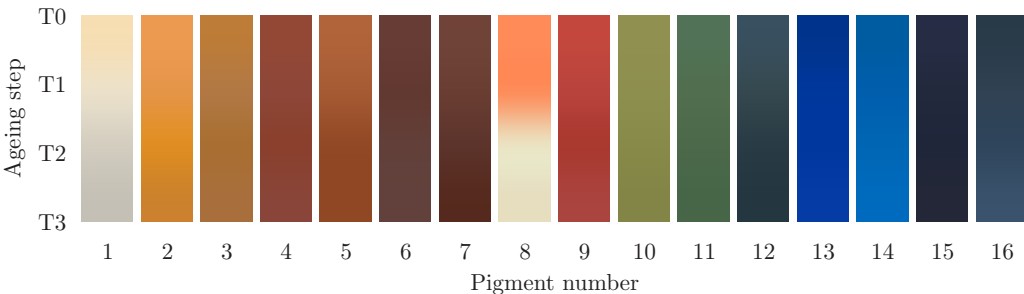

**Figure 5.** Color degradation after three aging steps for the fundamental pigments of Section 1.1.

Then, the quantification of color degradation due to the applied artificial aging is realized for each pigment separately. The lightness and saturation diagrams are demonstrated in Figure 6, while their exact values are summarized in Tables 2 and 3. Finally, Table 4 summarizes the $L^*$, $a^*$, and $b^*$ values of each pigment at the different time-cycles, and the color difference between successive steps is measured.

**Table 2.** Quantification of lightness at different aging steps for the fundamental pigments of Section 1.1. Values in parentheses denote the difference compared to the previous aging time-step.

| Pigment | Lightness | | | | | | |
| --- | --- | --- | --- | --- | --- | --- | --- |
| | T0 | T1 | | T2 | | T3 | |
| 1. Massicot | 72.78 | 72.21 | (−0.57) | 60.59 | (−11.62) | 52.49 | (−8.10) |
| 2. Naples yellow | 45.55 | 41.88 | (−3.67) | 38.03 | (−3.85) | 31.26 | (−6.77) |
| 3. Yellow ochre | 25.69 | 23.62 | (−2.07) | 20.61 | (−3.01) | 20.50 | (−0.11) |
| 4. Red ochre | 17.14 | 16.72 | (−1.42) | 15.72 | (−1.00) | 15.01 | (−0.71) |
| 5. Warm ochre | 23.34 | 20.09 | (−3.25) | 15.41 | (−4.68) | 14.84 | (−0.57) |
| 6. Hematite | 9.29 | 8.94 | (−0.35) | 8.74 | (−0.20) | 8.43 | (−0.31) |
| 7. Burnt Sienna | 9.99 | 8.67 | (−1.32) | 6.70 | (−1.97) | 5.34 | (−1.36) |
| 8. Minium | 47.58 | 48.21 | (+0.63) | 69.63 | (+21.42) | 71.51 | (+1.88) |
| 9. Cinnabar | 30.68 | 27.20 | (−3.48) | 23.06 | (−4.14) | 22.24 | (−0.82) |
| 10. Green earth | 20.98 | 20.18 | (−0.80) | 18.47 | (−1.71) | 16.96 | (−1.51) |
| 11. Malachite | 10.79 | 9.84 | (−0.95) | 8.74 | (−1.10) | 8.17 | (−0.57) |
| 12. Azurite | 7.40 | 5.76 | (−1.64) | 4.04 | (−1.72) | 3.38 | (−0.66) |
| 13. Ultramarine blue | 8.50 | 10.47 | (+1.97) | 10.87 | (+0.40) | 11.84 | (+0.97) |
| 14. Cobalt blue | 18.51 | 20.58 | (+2.07) | 23.11 | (+2.53) | 25.44 | (+2.33) |
| 15. Prussian blue | 3.18 | 2.65 | (−0.53) | 2.40 | (−0.25) | 2.38 | (−0.02) |
| 16. Indigo | 4.78 | 6.11 | (+1.33) | 6.88 | (+0.77) | 9.42 | (+2.54) |

**Table 3.** Quantification of saturation at different aging steps for the fundamental pigments of Section 1.1. Values in parentheses denote the difference compared to the previous aging time-step.

| Pigment | Saturation | | | | | | |
| --- | --- | --- | --- | --- | --- | --- | --- |
| | T0 | T1 | | T2 | | T3 | |
| 1. Massicot | 26.66 | 14.40 | (−12.26) | 9.32 | (−5.08) | 8.00 | (−1.32) |
| 2. Naples yellow | 66.11 | 68.85 | (+2.74) | 75.35 | (+6.50) | 78.91 | (+3.56) |
| 3. Yellow ochre | 69.05 | 68.68 | (−0.37) | 62.67 | (−6.01) | 61.43 | (−1.24) |
| 4. Red ochre | 63.60 | 60.59 | (−3.01) | 56.88 | (−3.71) | 54.33 | (−2.55) |
| 5. Warm ochre | 65.72 | 66.79 | (+1.07) | 71.60 | (+4.81) | 71.66 | (+0.06) |
| 6. Hematite | 41.34 | 39.46 | (−1.88) | 32.82 | (−6.64) | 29.85 | (−2.97) |
| 7. Burnt Sienna | 43.39 | 45.65 | (+2.26) | 46.59 | (+0.94) | 57.20 | (+10.61) |
| 8. Minium | 64.90 | 62.84 | (−2.06) | 18.93 | (−43.91) | 17.85 | (−1.08) |
| 9. Cinnabar | 69.39 | 66.78 | (−2.61) | 65.16 | (−1.62) | 61.00 | (−4.16) |
| 10. Green earth | 48.08 | 48.31 | (+0.23) | 49.22 | (+0.91) | 49.69 | (+0.47) |
| 11. Malachite | 15.26 | 20.27 | (+5.01) | 21.07 | (+0.80) | 21.11 | (+0.04) |
| 12. Azurite | 27.09 | 26.44 | (−0.65) | 24.07 | (−2.37) | 20.23 | (−3.84) |
| 13. Ultramarine blue | 81.88 | 82.38 | (+0.50) | 83.23 | (+0.85) | 83.75 | (+0.52) |
| 14. Cobalt blue | 74.16 | 71.82 | (−2.34) | 71.24 | (−0.58) | 70.85 | (−0.39) |
| 15. Prussian blue | 35.28 | 34.86 | (−0.42) | 33.95 | (−0.91) | 27.20 | (−6.75) |
| 16. Indigo | 29.30 | 29.35 | (+0.05) | 34.25 | (+4.90) | 35.27 | (+1.02) |

1.  Massicot is one of the two pigments showing a severe degradation in Figure 5, and it is confirmed by the quantification diagrams. Specifically, the lightness is almost not affected after the first aging step but a significant reduction is observed at the second and third ones. Although the saturation is decreased considerably at every step, the rate is limited until T3 instance. These results are validated by the color difference investigation since the alteration is severe at T1, but it tends to a smoother behavior as the aging progresses.
2.  The lightness of Naples yellow is fairly reduced especially at the last aging step, while the saturation is always at high levels and it is increasing at every step with a

noticeable change at T2. The color difference proves that, at the first aging step, there is a slight alteration, while a significant difference is observed at T2, due to saturation change, and at T3, due to lightness variation.

3.  Yellow ochre is generally darker than Naples yellow, and its lightness reduction is observed at the first two aging steps since at the third the alteration is only slight. The saturation is maintained at T1, but it is significantly decreased as the aging progresses, especially at T2. The color difference is observable at every time-step but mainly in the first aging step.

4.  The color stability of red ochre is remarkable regarding its lightness since a slight reduction is observed that tends to stabilize at the last aging steps. Nevertheless, the reduction of saturation is clear almost equally for every time-step. The color difference proves that the red ochre is quite stable since the alteration is noticeable mainly at T3.

5.  Another pigment that darkens is warm ochre since its already low lightness value is further decreased as the aging progress, especially at T2. On the contrary, its saturation is maintained in relatively high values and it is increased, mainly after the second step, while the color seems stabilized at T3. This observation is validated through the color difference since a noticeable alteration is observed at T2 in contrast to the insignificant one at T3.

6.  Hematite is a pigment of very low lightness that is almost not affected by aging since its overall alteration is the lowest regarding the investigated pigments. However, the saturation presents a decreasing tendency, while this effect is severe mainly after the second time step; thus, a more grayish tone is observed. This moderate alteration is observed in color difference, but the effect is more obvious in T1 step.

7.  The lightness of burnt Sienna is very low at the initial condition, and it is slightly dropped at every aging step. A completely different behavior is noted for the saturation that is fairly rising at T1; it seems unaffected at T2, and a significant increment is observed at T3. The color difference throughout aging procedure indicates that the alteration is augmented as the exposure is continued.

8.  The other pigment that presents severe degradation is minium. Specifically, although the first time-step seems to influence the pigment's color only slightly, a complete differentiation occurs on the second step. The initial moderate lightness is significantly increased at this step, while the relatively high saturation degrades to low values, therefore, tending to white. At the final step, the color is retained both in lightness and saturation. This behavior is confirmed by color difference since, at the second aging step, the most significant color transformation is noted.

9.  The intense color of cinnabar is degraded via aging since both lightness and saturation are decreased. Specifically, the lightness is influenced mainly at the first aging steps, while saturation drops significantly at T3. Consequently, a darker and less intense red is observed after the final aging procedure. This color alteration is observed in color difference values, since there is a change at every time-step, especially at T3 due to the saturation reduction.

10.  It is already stated that green earth is considered the most stable of pigments, and this is clarified via its saturation observation that is almost equivalent for any of the time-steps. Moreover, the lightness is reduced only by a slight factor, particularly after the second aging step. The color stability of green earth is clarified also via color difference values since they are less than the observable limit (2.3) even though the alteration is slightly increased after T2 due to lightness reduction.

11.  Malachite is another very stable pigment in terms of its color since the lightness is nearly unaffected by aging. Specifically, malachite is less luminous by a negligible factor, especially at T2. However, saturation is somewhat increased after the first time-step but is retained as the aging procedure continues. Color difference validates our results considering that the alteration is observable only at T1.

12. The counterpart to malachite is azurite that is generally darker, and its lightness is further decreased at the first aging steps while it is almost unaffected at the final step. Additionally, the saturation is stable at T1 but it is reduced as the aging progresses in contrast to malachite. The color difference indicates that the alteration is more evident at the initial stages of aging.

13. The ultramarine blue is a really intense color, as confirmed by the very high saturation values that are influenced only slightly by aging since a negligible increment is observed. Moreover, it is relatively dark; it gets somewhat brighter at T1, while its lightness is further increased at T2 and T3 by a very small factor. The color stability of ultramarine blue is verified by its color difference since the only observable change is occurred at T1.

14. An additional pigment of intense color is cobalt blue, presenting high saturation values of a fair reduction after the first aging step and insignificant counterparts for the remaining steps. However, the pigment becomes even brighter since its lightness is increased to moderate values at almost equivalent increments during aging. The color difference indicates that the aforementioned saturation change results in a moderate color alteration at T1, while the differentiation is decreased as aging progresses and the pigment tends to stabilize in terms of its color.

15. The darkest pigment of the panel, namely Prussian blue, presents the lowest lightness that it is affected negligibly by the aging process and particularly at first two aging steps. At these stages, the saturation is almost retained, while a considerable reduction is observed at the last time-step leading the pigment color to black. Finally, the color difference values indicate a relatively stable pigment that its alteration is augmented through time.

16. Indigo is an additional dark pigment but the aging procedure increases both its lightness and saturation; thus, a brighter color is perceived. Specifically, the lightness changes mainly at T1 and T3 steps, while a significant increment in saturation takes place in T2 and this value is further increased towards T3 via a small factor. The color difference clarifies the previous results since the alteration is always observable and it is boosted as the aging progresses.

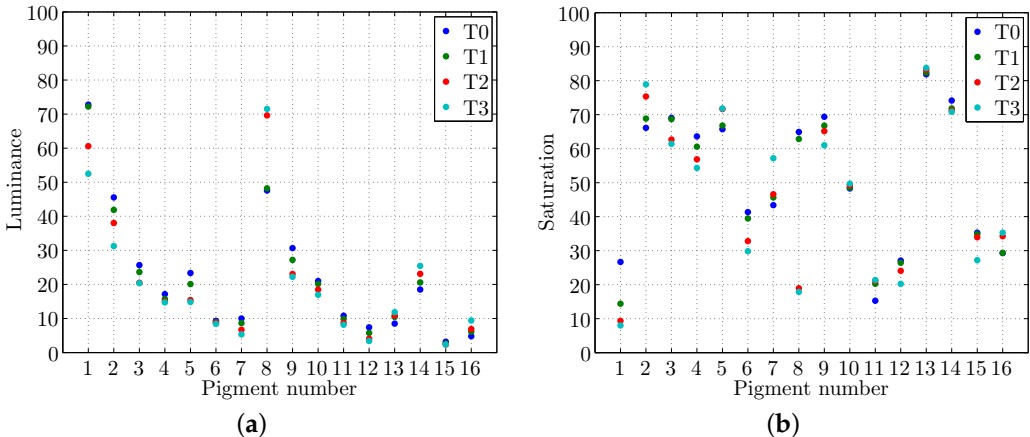

(**a**)  (**b**)

**Figure 6.** (**a**) Lightness and (**b**) saturation at different aging steps for the fundamental pigments of Section 1.1.

**Table 4.** $L^*$, $a^*$, and $b^*$ values of CIELAB 1976 at different aging steps for the fundamental pigments of Section 1.1. $\Delta E$ denotes the color difference compared to the previous aging time-step.

| | CIELAB 1976 | | | | | | | |
|---|---|---|---|---|---|---|---|---|
| Pigment | T0 | T1 | | T2 | | T3 | |
| | $L^*$, $a^*$, $b^*$ | $L^*$, $a^*$, $b^*$ | $\Delta E$ | $L^*$, $a^*$, $b^*$ | $\Delta E$ | $L^*$, $a^*$, $b^*$ | $\Delta E$ |
| 1. Massicot | 90, 4, 24 | 90, 1, 13 | 11.3 | 83, 1, 8 | 8.4 | 78, 1, 6 | 4.9 |
| 2. Naples yellow | 72, 27, 52 | 70, 26, 54 | 2.8 | 68, 27, 64 | 10.8 | 62, 25, 56 | 10.1 |
| 3. Yellow ochre | 59, 22, 47 | 57, 20, 39 | 8.7 | 53, 20, 43 | 5.4 | 53, 20, 38 | 5.2 |
| 4. Red ochre | 42, 31, 28 | 40, 30, 25 | 3.8 | 39, 29, 23 | 2.3 | 38, 31, 28 | 5.8 |
| 5. Warm ochre | 52, 29, 38 | 48, 29, 37 | 3.9 | 41, 30, 36 | 6.9 | 41, 30, 36 | 1.0 |
| 6. Hematite | 32, 15, 10 | 31, 19, 13 | 5.3 | 31, 16, 11 | 3.8 | 30, 19, 14 | 4.3 |
| 7. Burnt Sienna | 34, 20, 16 | 32, 20, 16 | 2.6 | 28, 18, 15 | 4.4 | 24, 20, 18 | 4.9 |
| 8. Minium | 72, 43, 49 | 73, 40, 47 | 3.4 | 89, −1, 16 | 53.3 | 90, −1, 18 | 1.9 |
| 9. Cinnabar | 50, 50, 36 | 48, 48, 33 | 4.2 | 43, 46, 34 | 4.7 | 45, 43, 29 | 6.3 |
| 10. Green earth | 59, −8, 34 | 58, −9, 33 | 1.2 | 56, −9, 33 | 1.9 | 54, −9,33 | 2.1 |
| 11. Malachite | 44, −19, 11 | 43, −17, 14 | 3.4 | 41, −17, 13 | 2.2 | 40, −18, 13 | 1.5 |
| 12. Azurite | 32, −6, −12 | 29, −7, −7 | 5.9 | 24, −6, −8 | 5.3 | 21, −6, −9 | 2.5 |
| 13. Ultramarine blue | 24, 20, −58 | 26, 22, −60 | 3.4 | 26, 23, −62 | 1.9 | 28, 22, −62 | 1.9 |
| 14. Cobalt blue | 36, −5 ,−47 | 37, −2, −52 | 6.4 | 40, −3, −51 | 3.4 | 42, −3, −53 | 2.7 |
| 15. Prussian blue | 18, 3, −16 | 17, 2, −14 | 2.7 | 16, 2, −11 | 3.1 | 15, 3, −14 | 3.3 |
| 16. Indigo | 23, −4, −12 | 27, −4, −13 | 3.7 | 28, −3, −17 | 4.0 | 33, −4, −18 | 5.2 |

*Comparison to State-of-the-Art Results*

Our analysis is finalized via the comparison of the evaluated discoloration characteristics to the reported ones in the literature. Initially, a very interesting colour stability analysis highlighted some aspects that are similar to our results [43]. In particular, the discussed stability of green earth pigment is validated, while a significant darkening in massicot, similar to our results, is reported. Moreover, the ochres are considered relatively stable, but certain conditions, such as the ones in our work, are able to degrade their color. Finally, there is also a validation considering the stability of malachite compared its blue counterpart azurite.

Additionally, there our various works that focus on specific egg-tempera pigments. In [44], various types of cinnabar were investigated and it confirms that the artificial aging decreases the lightness of the colour, while small fluctuations were observed in saturation. A similar work was conducted in [45], with a major difference that azurite was examined. Again, the artificial aging procedure in this paper justify the saturation degradation as well as an increased color difference. The authors in [46] presented their results concerning red ochre, where the lightness decreased, leading to a notable color difference. In our results, both saturation and lightness of red ochre decreased, indicating a good match to the literature. Another work [47] investigated various pigments under different aging procedures, revealing moderate color differences for azurite, cinnabar, naples yellow, and indigo, especially for thermal aging tests. It is worth mentioning that the time of exposure in this work is relatively low, so compared to our T1 results, the modereate color change is validated. Finally, UV LED-based aging was proposed in [48] for a small time-cycle of approximately 10 days. Consequently, the comparison conducted with our T1 results indicates a good match since the color difference is relatively low in both works.

## 4. Conclusions

An artificial aging procedure on Byzantine art egg-tempera pigments was conducted in this work in order to characterize the color degradation through different time-steps. A multi-material palette that includes various popular pigments was fabricated, and UV/Vis measurements were conducted at every individual aging step. The acquired spectra were further processed effectively to quantify the pigment color degradation via the examination of perceptual lightness and saturation. Several interesting results were extracted since a severe degradation was observed in a couple of pigments (massicot

and minium), while five pigments appeared stable through the aging process (red ochre, hematite, green earth, malachite, and Prussian blue). Finally, a moderate color darkening were detected in six pigments (Naples yellow, yellow and warm ochre, burnt Sienna, cinnabar, and azurite), while three blue pigments presented a tendency to discolor (ultramarine blue, cobalt blue, and indigo).

**Author Contributions:** Conceptualization, S.A. and G.K.; methodology, S.A.; measurements and validation, S.A. and G.A.; formal analysis, S.A.; investigation, S.A.; writing—original draft preparation, S.A.; writing—review and editing, S.A. and G.K.; visualization, S.A.; funding acquisition, G.K. All authors have read and agreed to the published version of the manuscript.

**Funding:** This work is part of the Iperion HS project that received funding from the European Union Horizon 2020 Framework Programme under grant agreement no. 871034 and "HELLAS-CH" project (MIS 5002735), which is implemented under the "Action for Strengthening Research and Innovation Infrastructures", funded by the Operational Programme "Competitiveness, Entrepreneurship, and Innovation" (NSRF 2014-2020) and co-financed by Greece and the European Union (European Regional Development Fund).

**Institutional Review Board Statement:** Not applicable.

**Informed Consent Statement:** Not applicable.

**Data Availability Statement:** The data presented in this study are available on request from the corresponding author. The data are not publicly available due to privacy.

**Conflicts of Interest:** The authors declare no conflicts of interest.

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
