# Peer review of "Consistent Characterization of Color Degradation Due to Artificial Aging Procedures at Popular Pigments of Byzantine Iconography"

_minerals, doi:10.3390/min11070782_

Round 1

Reviewer 1 Report

The paper was greatly improved since the last revision. A thorough revision of the English language is needed before publication.

Reviewer 2 Report

with the revisions done by the authors I think that the manuscript can be published

This manuscript is a resubmission of an earlier submission. The following is a list of the peer review reports and author responses from that submission.

Round 1

Reviewer 1 Report

The manuscript presents what could be an interesting work on the topic of pigments degradation. Nevertheless it is not clear what is the novelty of the present work, since there is an extensive bibliography on the topic of degradation of these pigments.

Furthermore, relevant information on the experimental details is missing, such us the origin of the studied pigments.

The introduction and discussion of the results need a major revision Most of the information can be obtained in the already existent published bibliography.

Author Response

The manuscript presents what could be an interesting work on the topic of pigments degradation.”

We would like to thank the reviewer for his/her valuable comments that really enhanced the output of our paper.

(1) Nevertheless it is not clear what is the novelty of the present work, since there is an extensive bibliography on the topic of degradation of these pigments.”

We applied several revisions in order to highlight the novelty of our work that is not the comprehensive study of the chemical effect of discoloration but the thorough color characterization throughout equivalent aging time-cycles. Additionally, our analysis is not limited in the study of color alteration in a single instance but we want to determine the color stability, the type of color degradation and the dynamic change of the latter during the aging steps, namely whether it is observed at early of later stages of aging [Abstract, page 1, sentences 9-11 / Introduction, page 2, sentences 61-65 / Results and Discussion, page 8, sentences 276-286].

(2) Furthermore, relevant information on the experimental details is missing, such us the origin of the studied pigments.”

The studied pigments have been provided by Kremer Pigments Inc. and this is now included in the paper [Materials and Methods, page 3, sentence 104].

(3) “The introduction and discussion of the results need a major revision Most of the information can be obtained in the already existent published bibliography.”

As explained in the previous comment, we intend to go further than the color degradation at a single. We are achieving this via the observation at the different time-cycles in order to evaluate the dynamic color change during the exposure [Abstract, page 1, sentences 9-11 / Introduction, page 2, sentences 61-65 / Results and Discussion, page 8, sentences 276-286]. Moreover, we enhanced considerably the Results and Discussion section in order to highlight our scope [Results and Discussion, pages 10-13, sentences 321-402].

Reviewer 2 Report

The manuscript report a systematic study of the effect of ageing on the pigment palette of Byzantine iconography. Generally, the context of pigment alteration in the Byzantine paintings is poorly reported, and this make less relevant the correlation of the obtained results with specific problems of conservation/restoration. No mention is reported of numerous studies devoted to the study of alteration due to ageing of most of the reported pigments, and some of them with significant physico-chemical interpretation of pigment alteration mechanism. The value of this research is the systematic approach, but the discussion of results sounds sketchy.

Some doubtful point:

line 24: the perception of color is the fundamental  interaction of human with artworks, this sentence is questionable

line 64: the difference between moisture and humidity is not clear

line 110: Goethite formula is FeO(OH)

line 171: a better and more systematic measurement of environmental condition could be done, and discussed in relation with the pigment ageing

Figure 3 (b), caption : scale axis lacks of indications (X,Y ?)

Author Response

The manuscript report a systematic study of the effect of ageing on the pigment palette of Byzantine iconography. The value of this research is the systematic approach, but the discussion of results sounds sketchy.”

We would like to thank the reviewer for his/her recommendations that really helped to improve our study.

(1) Generally, the context of pigment alteration in the Byzantine paintings is poorly reported, and this make less relevant the correlation of the obtained results with specific problems of conservation/restoration. No mention is reported of numerous studies devoted to the study of alteration due to ageing of most of the reported pigments, and some of them with significant physico-chemical interpretation of pigment alteration mechanism.”

We included a couple of studies regarding the reported psysiochemical degradation in Byzantine paintings [References 29, 30]. Moreover, we applied several revisions in order to highlight the novelty of our work that is not the comprehensive study of the chemical effect of discoloration but the thorough color characterization throughout equivalent aging time-cycles. Additionally, our analysis is not limited in the study of color alteration in a single instance but we want to determine the color stability, the type of color degradation and the dynamic change of the latter during the aging steps, namely whether it is observed at early of later stages of aging [Abstract, page 1, sentences 9-11 / Introduction, page 2, sentences 61-65 / Results and Discussion, page 8, sentences 276-286].

(2) the perception of color is the fundamental  interaction of human with artworks, this sentence is questionable”

We rephrased this specific sentence by replacing the word "fundamental" to "initial" since we intend to mean that the vision is the first interaction with traditional artworks or icons.

(3) “the difference between moisture and humidity is not clear”

There was a mistyping in the previous version of the paper since we are not implying any difference between moisture and humidity. This error is now corrected.

(4) Goethite formula is FeO(OH)”

We removed the formula since it is not providing any added value.

(5) a better and more systematic measurement of environmental condition could be done, and discussed in relation with the pigment ageing”

Additional details regarding the environmental conditions have been included. Specifically, the Mediterranean area ones are now described thoroughly [Materials and Methods, page 5, sentences 201-205].

(6) Figure 3 (b), caption : scale axis lacks of indications (X,Y ?)”

The axis labels in Figure 3b have been included [Figure 3, page 7].

Reviewer 3 Report

This paper analyses the changes in colour of a number of egg tempera paints exposed to accelerated aging. Degradation of pigments due to environmental factors – e.g relative humidity, temperature, illumination, pollutants – has been extensively investigated in literature. This study aims at characterizing the colour-degradation of pigments by means of the variation in luminance and saturation obtained by UV-Vis reflectance spectroscopy. The artificial aging simulates the environmental fluctuations of historical buildings, in which byzantine egg-tempera paints are typically conserved. Despite the clarity of the text, several issues should be addressed to improve the scientific relevance and novelty of this study.

Major correction:

In my opinion, the analysis should involve more quantitative measurements and extended data analysis, in order to provide a reliable characterization of colour changes. Besides, the parameters used for the characterization (luminance and saturation) are not sufficient for defining variations in hue, fading, discoloration, darkening. A multi-analytical approach, including other complementary non-invasive techniques, would be a more suitable choice. For example, colorimetry could be used to highlight change in colour after the ageing treatment. Raman and laser-induced fluorescence spectroscopy could be applied to investigate the chemical effect of the discoloration, especially in the binding medium. Fourier transform infrared spectroscopy would allow identifying specific differences in absorption, which would turn useful for explaining the different behaviour of the paints.

Furthermore, I have some doubts on the calculation used to obtain both the saturation and the luminance parameters. It is not clear to me how the changes in saturation reported in Fig. 5 were assessed: is it just a qualitative representation of what observed?

Concerning equation (1), could the authors please supply some references? It seems that L has the same dimensions as a wavelength.

At line 203 it is stated that “The described procedure is briefly illustrated in Figure 3a for a cobalt blue sample”: this is not straightforward for the reviewer. The spectrum of cobalt blue is reported with the visible region enhanced, which is trivial for colorimetric computation, but does not explain the procedure.

Moreover, the unit of measurement (percentage?) are missing (both in fig. 3a and 3b), and the error bar of luminance and saturation values should be reported in tab. 2 and in fig. 5 a and b: the differences among values measured at each aging steps could be significant or negligible, depending on the error. 

Changes in luminance and colour are traditionally quantified with the colorimetric coordinates (L*, a*, b*) and colour difference (DE*) of the CIELab colour space. Is there any reason for using HSL coordinates?

Artificial Aging Procedure: what is the correspondence between natural aging time and the 4 steps of accelerated aging? Is it possible to quantify it in terms of years of natural exposure? What was the irradiance (W/cm2)?

Materials and Methods, line 73-76: the authors report that “according to the Byzantine iconography, successive layers of lightings and writings are applied, by adding white lead or carbon black/red ochre, respectively, in the same mixture of initial color. Although this procedure is adopted in various samples of our panel, as depicted in Figure 1, this paper focuses only on pure colors, namely without any lighting or writing.” Why this choice? The analysis of the more realistic paints, with lighting and varnish, would have made the study more complete and interesting.

In the description of the instrument (spectrophotometer @line 189-190) it is stated the following: “A monochromator (prism or grating) analyses the white light in different monochromatic areas and selects the desired wavelength with high accuracy”: do you have the choice of switching between the two dispersion elements? In case yes, could you please specify which one is used?

At line 200-201 is stated “Specifically, the objectivity of the color measurement is ensured via a calibration procedure of the maximum and minimum reflection values utilizing white barium sulfate and carbon black, respectively”. Why in eq. (1) measurements on carbon black does not appear?

Minor correction:

Line 37-38: in more controlled conditions than those found in natural conditions

Line 39: No comma after “Although”

Line 46: processes that paintings undergo within months to years from their realization

Line 60: the degradation due to artificial aging of egg-tempera pigments used in Byzantine art.

Line 62: namely without the presence of protective varnishes, lightings or writings.

Line 64: historical buildings, in which Byzantine artworks are commonly conserved.

Line 94: please add here a reference to figure 1, otherwise the connection between the list of pigments and the examined palette would not be straightforward to the readers 

Line 107: its color is due to the iron oxide

Line 118: The pigment presents/is characterized by a rich dark color with purple-red hue

Line 119: as they present a form of bright elongated alder. You mean a branched form?

Line 137: as the most permanent of all pigments

Line 253: Massicot is one of the two pigments showing a severe degradation/in which a severe a severe degradation was observed

Author Response

This paper analyses the changes in colour of a number of egg tempera paints exposed to accelerated aging. Degradation of pigments due to environmental factors – e.g relative humidity, temperature, illumination, pollutants – has been extensively investigated in literature. This study aims at characterizing the colour-degradation of pigments by means of the variation in luminance and saturation obtained by UV-Vis reflectance spectroscopy. The artificial aging simulates the environmental fluctuations of historical buildings, in which byzantine egg-tempera paints are typically conserved.”

We would like to thank the reviewer for his/her extensive and careful study of our paper and the very positive attitude and respect for our work.

(1) In my opinion, the analysis should involve more quantitative measurements and extended data analysis, in order to provide a reliable characterization of colour changes. Besides, the parameters used for the characterization (luminance and saturation) are not sufficient for defining variations in hue, fading, discoloration, darkening. A multi-analytical approach, including other complementary non-invasive techniques, would be a more suitable choice. For example, colorimetry could be used to highlight change in colour after the ageing treatment. Raman and laser-induced fluorescence spectroscopy could be applied to investigate the chemical effect of the discoloration, especially in the binding medium. Fourier transform infrared spectroscopy would allow identifying specific differences in absorption, which would turn useful for explaining the different behaviour of the paints.”

The effect of the egg-yolk binder is not neglected and we provided some indicative FTIR results [Figure 4, page 8] that prove the binder degradation due to artificial aging procedure. However, the chemical effect of discoloration is out of scope of this work since we focus explicitly on the color characterization throughout equivalent aging time-cycles in order to determine the color stability, the type of color degradation and the time instance of the latter, namely whether it is observed at early of later stages of aging [Abstract, page 1, sentences 9-11 / Introduction, page 2, sentences 61-65 / Results and Discussion, page 8, sentences 276-286]. Moreover, we utilized UV/Vis spectroscopy due to its superiority compared to colorimetry as discussed, now, in the text [Materials and Methods, page 7, sentences 240-243 / Results and Discussion, page 8, sentences 285-286].

(2) Furthermore, I have some doubts on the calculation used to obtain both the saturation and the luminance parameters. It is not clear to me how the changes in saturation reported in Fig. 5 were assessed: is it just a qualitative representation of what observed?”

The proposed calculation of luminance and saturation is novel in this work since we are relying on the rich in information spectral data. These values are a percentage of the maximum value, namely the totally reflecting white for luminance and a spectral color for saturation. Considering the latter, it is measure of how saturated is a color, namely the relative distance from the equivalent spectral one [Materials and Methods, page 7, sentences 240-243 / Results and Discussion, pages 7-8, sentences 261-264].

(3) “Concerning equation (1), could the authors please supply some references? It seems that L has the same dimensions as a wavelength.”

In this equation L is an integration of the ratio between the measured spectrum to the totally reflecting one. Consequently L is dimensionless. The integration is now separated to nominator and denominator in order to avoid this confusion [Equation 1, page 7].

(4) At line 203 it is stated that “The described procedure is briefly illustrated in Figure 3a for a cobalt blue sample”: this is not straightforward for the reviewer. The spectrum of cobalt blue is reported with the visible region enhanced, which is trivial for colorimetric computation, but does not explain the procedure.”

In this Figure, the highlighted area corresponds to the integration for the luminance extraction [Materials and Methods, page 7, sentences 251-252].

(5) Moreover, the unit of measurement (percentage?) are missing (both in fig. 3a and 3b), and the error bar of luminance and saturation values should be reported in tab. 2 and in fig. 5 a and b: the differences among values measured at each aging steps could be significant or negligible, depending on the error.”

The luminance and saturation values are dimensionless, so we are, now, representing them as a percentage of their maximum value (explained in comment 2) and the Figures are changed accordingly [Figure 7, page 10]. Moreover, the initial table with the extracted values is now split into two different regarding luminance and saturation. These new tables are, now, including the difference between the aging steps [Table 2, page 10 / Table 3, page 11].

(6) Changes in luminance and colour are traditionally quantified with the colorimetric coordinates (L*, a*, b*) and colour difference (DE*) of the CIELab colour space. Is there any reason for using HSL coordinates?”

First of all, we are not using conventional HSL coordinates since we are evaluating the luminance and saturation values directly via the spectrum. Moreover, the colorimetric coordinates (L*, a*, b*) the color difference calculation are now included as an auxiliary procedure [Materials and Methods, page 8, sentences 265-273], while an additional Table is now present and discussed [Table 4, page 12]. However, the color difference is not providing any information regarding the direction of color change. This is the main reason that luminance and saturation are calculated via the UV/Vis spectrum.

(7) Artificial Aging Procedure: what is the correspondence between natural aging time and the 4 steps of accelerated aging? Is it possible to quantify it in terms of years of natural exposure? What was the irradiance (W/cm2)?”

It was stated in the original version of the paper that the accurate quantification of artificial aging in terms of natural exposure is not possible, but we tried to emulate approximately 50 years via each aging step. Moreover, the irradiance has a mean value of almost 800 W/m2. These details are, now, explicitly noted in the paper [Materials and Methods, page 6, sentences 214-216].

(8) Materials and Methods, line 73-76: the authors report that “according to the Byzantine iconography, successive layers of lightings and writings are applied, by adding white lead or carbon black/red ochre, respectively, in the same mixture of initial color. Although this procedure is adopted in various samples of our panel, as depicted in Figure 1, this paper focuses only on pure colors, namely without any lighting or writing.” Why this choice? The analysis of the more realistic paints, with lighting and varnish, would have made the study more complete and interesting.”

The pigments with varnish, or lightings and writings, have been avoided in our work since we didn't wanted a multi-parametric discoloration effect because it could lead in more arbitrary conclusions. Consequently, this paper focuses on pure pigments in order to extract robust results [Materials and Methods, page 3, sentences 85-88].

(9) In the description of the instrument (spectrophotometer @line 189-190) it is stated the following: “A monochromator (prism or grating) analyses the white light in different monochromatic areas and selects the desired wavelength with high accuracy”: do you have the choice of switching between the two dispersion elements? In case yes, could you please specify which one is used?”

In this work we are using a grating monochromator due to its advantageous linear performance at a wide UV/Vis spectral range [Materials and Methods, page 6, sentences 225-227]

(10) At line 200-201 is stated “Specifically, the objectivity of the color measurement is ensured via a calibration procedure of the maximum and minimum reflection values utilizing white barium sulfate and carbon black, respectively”. Why in eq. (1) measurements on carbon black does not appear?”

The reviewer is correct and we have now changed this equation in order to take into consideration both white and black references [Equation 1, page 7].

(11) Minor corrections:[...]”

All minor corrections have been taken under consideration and have been successfully addressed.

Round 2

Reviewer 1 Report

The present paper was revised following previous comments. The discussion of the results was greatly improved, but it still lacks a discussion within the state of the art. There are several published work on the degradation, colour change, etc., of the studied pigments in egg tempera and the obtained results should be compared with the literature.

The text also needs a final revision, and I propose that the list o pigments and description of section 2.2.1. should pe part of the introduction. References used for the description of the pigments need to be added.

Reviewer 2 Report

Check accurately the English language, there are some typos in the text. The figure 4 and its comment in the manuscript, notwithstanding the binder aging is not a topic of the reported research, is not sufficiently explained.

Reviewer 3 Report

The authors have better focused the scope of the study and enriched the analysis by properly adding colorimetry data. However, the study still lacks of scientific robustness and I think it needs more time to improve it. See the uploaded file for further comments.
